



# Measurement report: Balloon-borne in-situ profiling of Saharan dust over Cyprus with the UCASS optical particle counter

Maria Kezoudi[1,2], Matthias Tesche[1,3], Helen Smith[1,4], Alexandra Tsekeri[5], Holger Baars[6], Maximilian Dollner[7], Víctor Estellés[8,9], Bernadett Weinzierl[7], Zbigniew Ulanowski[1,10,11], Detlef Müller[1], and Vassilis Amiridis[5]

[1]University of Hertfordshire, Hatfield, United Kingdom
[2]now at The Cyprus Institute, Nicosia, Cyprus
[3]now at Leipzig University, Leipzig, Germany
[4]now at TruLife Optics Ltd, London, United Kingdom
[5]National Observatory of Athens, Athens, Greece
[6]Leibniz Institute for Tropospheric Research, Leipzig, Germany
[7]Faculty of Physics, Aerosol Physics and Environmental Physics, University of Vienna, Vienna, Austria
[8]University of Valencia, Valencia, Spain
[9]ISAC-CNR, Rome, Italy
[10]School of Earth and Environmental Sciences, University of Manchester, Manchester, United Kingdom
[11]British Antarctic Survey, NERC, Cambridge, United Kingdom

**Correspondence:** Maria Kezoudi (m.kezoudi@cyi.ac.cy)

**Abstract.** This paper presents measurements of mineral dust concentration in the diameter range from 0.4 to 14.0 $\mu$m with a novel balloon-borne optical particle counter, the Universal Cloud and Aerosol Sounding System (UCASS). The balloon launches were coordinated with ground-based active and passive remote-sensing observations and airborne in-situ measurements with a research aircraft during a Saharan dust outbreak over Cyprus from 20 to 23 April 2017. The aerosol optical depth

at 500 nm reached values up to 0.5 during that event over Cyprus and particle number concentrations were as high as 50 cm$^{-3}$ for the diameter range between 0.8 and 13.9 $\mu$m. Comparisons of the total particle number concentration and the particle size distribution from two cases of balloon-borne measurements with aircraft observations show reasonable agreement in magnitude and shape despite slight mismatches in time and space. While column-integrated size distributions from balloon-borne measurements and ground-based remote sensing show similar coarse-mode peak concentrations and diameters, they illustrate the

ambiguity related to the missing vertical information in passive sun photometer observations. Extinction coefficient inferred from the balloon-borne measurements agrees with those derived from coinciding Raman lidar observations at height levels with particle number concentrations smaller than 10 cm$^{-3}$ for the diameter range from 0.8 to 13.9 $\mu$m. An overestimation of the extinction coefficient of a factor of two was found for layers with particle number concentrations that exceed 25 cm$^{-3}$. This is likely the result of a variation in the refractive index, the shape- and size-dependency of the extinction efficiency of dust

particles along the UCASS measurements.



## 1 Introduction

Atmospheric aerosols are of significant importance for the Earth's radiative budget. They have a direct impact on climate by scattering and absorbing solar radiation. They can also act as ice nucleating particles and cloud condensation nuclei, and thus, influence not only the formation and evolution of clouds but also the hydrological cycle (Stocker et al., 2013). Aerosols and

their precursors originate from natural and anthropogenic sources. Natural sources include emissions from the ocean, soils, volcanoes, and vegetation, whereas anthropogenic sources include emissions from the combustion of fossil fuels or the result of changes in land use (Boucher, 2015). For instance, sulphates and soot can be of both natural and anthropogenic origin, while mineral dust and marine aerosols originate predominantly from natural sources (Stocker et al., 2013). The latter two aerosol types are abundant in the atmosphere and particularly mineral dust can be transported over intercontinental distance from its

source regions (Prospero, 1999; Weinzierl et al., 2017).

Over the past 15 years, several measurement campaigns have focused on gaining deeper insight into the role of mineral dust on the Earth's system. An overview of several mineral dust field campaigns is given in Weinzierl et al. (2017). These experiments generally featured comprehensive remote-sensing instrumentation, detailed monitoring of chemical, microphysical, and optical properties of aerosols at the surface as well as airborne in-situ observations with research aircraft. Such observations

have been performed, for instance, during the two Saharan Mineral Dust Experiments (SAMUM, Weinzierl et al. 2009; Ansmann et al. 2011), Fennec (Ryder et al., 2013), the Saharan Aerosol Long-Range Transport and Aerosol–Cloud-Interaction Experiment (SALTRACE, Weinzierl et al. 2017), and the CHemistry and AeRosols Mediterranean EXperiments (CHArMEx, Renard et al. 2018). Recently, the focus of such activities has extended towards the eastern Mediterranean as this region is on the cross road of aerosol transport of mineral dust from Sahara and Middle East, continental outflow from Europe, as well

as biomass-burning smoke from eastern Europe and central Asia (Georgoulias et al., 2016). The majority of dust storms over the eastern Mediterranean basin occurs between December and April with maximum dust load during April (Israelevich et al., 2002). The main zones of cyclogenesis in the Mediterranean Sea determine dust uplift and transport in the region (Alpert et al., 1990). Heavy dust periods over the eastern Mediterranean are frequently associated with the so-called Cyprus Low (Katsnelson, 1970; Dayan et al., 2008) as well as the Sharav cyclone (Alpert and Ziv, 1989) which transport dust from the Arabian

deserts and northern Sahara into the eastern Mediterranean basin where they are frequently observed over Cyprus.

Statistical information on the size distributions of atmospheric aerosols, cloud droplets, and ice crystals is of vital importance for identifying and evaluating the physical processes governing aerosol-cloud interactions and their climate effects which currently contribute considerable uncertainty to our understanding of current and future climate change (Stocker et al., 2013) as well as to the performance of Numerical Weather Prediction models (Baldauf et al., 2011). The majority of the data assimilated

into models and used for model verification comes from remote-sensing observations (Lahoz and Schneider, 2014). Meteorological soundings in combination with an optical particle counter (OPC) can provide time series of aerosol size distribution profiles that have the potential to complement the data for assimilation in and verification of atmospheric models. The purpose of this paper is to present results of in-situ measurements of mineral dust particles over the eastern Mediterranean with a novel disposable balloon-borne OPC and to assess the quality of the collected data based on independent observations.





OPCs are well-established optical instruments for the measurement of particle size distributions in the size range between 0.060 and 100 μm. However, only few current OPCs have been specifically developed for balloon-borne measurements, which can only be performed with light-weight instruments. The Light Optical Aerosol Counter (LOAC) is a balloon-borne OPC that was designed for the detection of irregularly shaped aerosols in the diameter range from 0.2 to 100.0 μm (Renard et al., 2016). It was deployed for aerosol profiling during a dust event in the framework of CHArMEx in 2013 (Renard et al., 2018). The

non-disposable LOAC weighs about 1 kg and the sampled air is drawn through an inlet into the measurement chamber. Hence, the LOAC is not suitable for cloud sampling as cloud droplets might not be able to pass through the inlet without shattering or evaporation losses. The Cloud Particle Sensor (CPS, Fujiwara et al. 2016) is a balloon-borne instrument for measuring cloud particle number concentrations in the diameter range from 2 to 80 μm. It measures the state of polarisation of the scattered laser light and provides information on cloud phase. The CPS also employs an inlet sampling system. The CPS was developed

specifically for cloud measurements and the lower detection limit allows to sample only coarse aerosol particles. In contrast to conventional OPCs (with the exception of wing-mounted probes) that draw air through a narrow channel behind an inlet, the Universal Cloud and Aerosol Sounding System (UCASS) OPC was developed with an open sampling path (Smith et al., 2019). This open-path design reduces cut-off and shattering effects that can lead to counting and sizing uncertainties and makes the instrument suitable for measurements of both aerosols and clouds.

Measurements presented here have been performed in the framework of the European Research Council (ERC) project entitled Absorbing aerosol layers in a changing climate: aging, lifetime and dynamics (A-LIFE, https://www.a-life.at/) that was based on Cyprus. The aim of A-LIFE was to investigate the properties of absorbing aerosols, particularly of mixtures of mineral dust and black carbon. The activities incorporated measurements with a research aircraft, advanced aerosol lidars, sun photometers, and ground-based in-situ instrumentation. An intense dust outbreak from 20 to 23 April 2017 provided ideal

conditions for deploying the UCASS OPC for balloon-borne in-situ dust profiling. This paper is organised as follows. Section 2 presents the instrumentation and methods. Findings are described and discussed in Section 3. Conclusions and a summary are given in Section 4.

## 2  Instruments and methods

### 2.1  UCASS

The Universal Cloud and Aerosol Sounding System (UCASS) is a lightweight, disposable OPC that was developed at the University of Hertfordshire (Smith et al., 2019). The UCASS unit weighs 280 g and was designed for use as a balloon-borne instrument, as a dropsonde, or on an Unmanned Aerial Vehicle (UAV, Girdwood et al. 2020). The UCASS OPC features an open-path geometry that prevents particle losses and droplet shattering by the inlet that would need to be corrected for during data analysis (Smith et al., 2019). The instrument operates a 658-nm laser diode and collects light scattered by individual

particles in an angular range between 16° and 104°. Depending on the configuration mode and the laboratory calibration, UCASS can measure either aerosols in the diameter range between 0.4 and 17.0 μm or cloud droplets in the detection range from 1.0 to 40.0 μm. The uncertainty of the number concentration measured when UCASS is launched with a radiosonde was





found to be about 8% using computational fluid dynamics modelling as presented in Smith et al. (2019). This value results from varying airflow related to the tilt of the instrument. Comparisons to reference instruments during laboratory experiments with a fixated UCASS showed a smaller uncertainty of the measured number concentrations. A detailed description of the instrument and its calibration can be found in Smith et al. (2019).

The UCASS OPC is typically deployed in combination with a Graw DFM-09 radiosonde (https://graw.de/products/radiosondes/dfm-09) which is used to measure relative humidity, temperature and pressure. At every 1 s interval, the dataset is either saved to an on-board micro SD card or transmitted via a serial link (XDATA protocol) to a radiosonde device for radio frequency transmission of the data (Smith et al., 2019). The DFM-09's XDATA interface is used for transmitting the UCASS data in 10 size bins together with time of flight data for quality assurance. Other sondes employing the XDATA protocol can be used as well. The UCASS-radiosonde payload can be used to obtain aerosol and cloud profiles from the surface up to the tropopause within about 60 min from launch. The flow speed through the UCASS' open detection path is determined by the ascent rate $u$ of the meteorological balloon. Effects of a tilt of the instrument on the flow rate are discussed in Smith et al. (2019). During the launch preparation, the balloon is filled to a size that translates into an ascent rate of about $5\,\mathrm{ms}^{-1}$ to guarantee optimum measurement performance of the UCASS. In the data analysis, the ascent rate is calculated from the change in height $h$ with time $t$ by $u = \Delta h/\Delta t$. The ascent rate is used to calculate the volume $v$ of sampled air by $v = Aut$ with the UCASS sample area of $A = 5.0 \times 10^{-7}\,\mathrm{m}^2$ which is specified as a section of the laser beam (Smith et al., 2019). The device electronics can measure up to 104 particles per second and can operate in air flow speeds between 2 and 15,$\mathrm{ms}^{-1}$, with the standard firmware. For a standard operating velocity of $5\,\mathrm{ms}^{-1}$, the corresponding particle concentration is $3.5 \times 10^9\,\mathrm{m}^{-3}$ (Smith et al., 2019).

The raw particle counts $C$ per size bin $i$ are used to calculate the particle number concentration per size bin $n_i = C_i/V$ as number of particles per unit volume over the covered size range. Summation of $n_i$ over all size bins leads to the total number concentration $N$. The particle number size distribution is determined by

$$\mathrm{d}n_i/\mathrm{dlog}D_i = \frac{n_i}{\log D_{i+1} - \log D_i} \tag{1}$$

with the assumption of spherical particles. While mineral dust particles are non-spherical, the shape effect on the scattering phase function with respect to spherical particles is less pronounced within the angular range exploited in the UCASS setup (forward to sideward scattering) compared to scattering in the backward direction. Hence, the use of Mie scattering has a small effect on the calculated size distributions even in the presence of non-spherical particles (Johnson and Osborne, 2011; Lacis and Mishchenko, 1995).

The column-integrated volume size distribution for comparison to the normalised volume size distributions provided from remote-sensing retrievals is calculated using the sum of the number concentration for each bin over the entire ascent together with the bin centre $(D_{i+1} + D_i)/2$ and width $D_{i+1} - D_i$ by

$$\mathrm{d}V_i/\mathrm{dlog}D_i = \frac{\pi n_i}{6} \frac{\left(\frac{D_{i+1}+D_i}{2}\right)^3}{\log D_{i+1} - \log D_i}. \tag{2}$$





The effective diameter is defined by Hansen (1971) as the ratio of the volume to the surface-area concentration by

$$d_{\text{eff}} = \frac{\int_0^\infty n(D_{\text{i}}) D_{\text{i}}{}^3 dD_{\text{i}}}{\int_0^\infty n(D_{\text{i}}) D_{\text{i}}{}^2 dD_{\text{i}}}. \tag{3}$$

UCASS measurements can be used to calculate the aerosol extinction coefficient. This can then be compared to the extinction coefficient profile derived from collocated lidar measurements. For a particle with diameter $D$ and known refractive index, the size-dependent extinction efficiency $Q_{\text{ext}}(D)$ (unitless) can be derived from Mie-scattering calculations. Here, we use a refractive index of 1.52+0.002i. Then, the extinction cross section of the particle (in m$^2$) is calculated by $C_{\text{ext}} = (\pi D^2/4) Q_{\text{ext}}(D)$.
Using the measured number concentration, the extinction coefficient (in m$^{-1}$) is derived by

$$\alpha = \sum_{i=1}^{10} n_i C_{\text{ext},i}. \tag{4}$$

## 2.2 A-LIFE Instrumentation

In order to demonstrate the UCASS' capability for profiling of aerosol number concentrations and size distributions, the quality of its observations needs to be evaluated with the help of independent data. To meet optimum conditions for comparison,
UCASS launches during A-LIFE were coordinated with ground-based remote sensing (ensuring also the temporal collocation of active and passive instruments) and the flight schedule of the DLR-Falcon research aircraft.

A Polly$^{\text{XT}}$ multiwavelength aerosol Raman lidar (Engelmann et al., 2016) from the Institute for Tropospheric Research (TROPOS), Leipzig, Germany, was operated at Limassol from October 2016 to March 2018. Polly$^{\text{XT}}$ measures profiles of aerosol backscatter coefficients at 355, 532, and 1064 nm, aerosol extinction coefficients at 355 and 532 nm, and aerosol
linear depolarisation ratios at 355 and 532 nm. These measurements provide insight into the vertical distribution of aerosol concentration, size, and type (Engelmann et al., 2016). Near real-time data from the Polly$^{\text{XT}}$ website (http://polly.tropos.de) were consulted to schedule UCASS launches for dust observations.

Aerosol Robotic Network (AERONET) sun photometer measurements (Holben et al., 1998) during A-LIFE were performed at Paphos and Limassol. These measurements provide information on the optical and microphysical properties of the bulk
aerosol in the atmospheric column. AERONET sun photometers perform spectrally resolved measurements of aerosol optical depth (AOD) at 340, 440, 675, 870, 1020, and 1640 nm and of sky radiances at several almucantar angles at 440, 675, 870, and 1020 nm (Holben et al., 1998). In this work, only AERONET version 3 level 2.0 data are considered.

The DLR-Falcon research aircraft was equipped with an extensive in-situ aerosol payload including total aerosol concentration measurements (0.005 - 930 µm), highly resolved size distribution measurements in the range between 0.25 and 930 µm
particle diameter, a wind lidar and meteorological sensors. Furthermore, aerosol optical properties were determined, and particles were collected for offline chemical analyses. The setup was similar to earlier campaigns that also focused on mineral dust (Weinzierl et al., 2009, 2011, 2017). Local column closure flights were performed at the sites of Paphos airport and the Limassol lidar station. In this paper, UCASS measurements are compared to data collected with a second generation Cloud, Aerosol, and Precipitation Spectrometer (CAPS, Spanu et al. 2020) that was mounted at the aircraft wing. The CAPS instrument consists
of a Cloud and Aerosol Spectrometer with Depolarization Detection (CAS), and a Cloud Imaging Probe (CIP). Furthermore,





it contains a few minor sensors including a Liquid Water Content sensor (LWC), a pitot tube measuring the airspeed and sensors for pressure, temperature and relative humidity. The CAS uses a 658 nm laser to observe the size distribution of particles between approx. 0.5 and 50.0 μm. The CIP uses a linear array of 64 photodiodes to detect shadow images of particles in the size range between 15 and 930 μm in diameter. For the comparison to UCASS observations, CAPS measurements were opted

to overlap with the UCASS sampling range from 0.79 to 13.90 μm in diameter. During A-LIFE, a total of 17 research flights were performed over the eastern Mediterranean, i.e. from or to Paphos. Two of those flights could be matched to UCASS measurements in both time and space.

## 2.3 Remote sensing retrievals

UCASS in-situ measurements of the particle size distribution and the subsequently derived extinction coefficient are also

evaluated with the findings from remote-sensing observations. For this purpose, lidar and sun-photometer data are used as input to the Generalised Aerosol Retrieval from Radiometer and Lidar Combined data algorithm (GARRLiC, Lopatin et al. 2013) and the AERONET (Dubovik et al., 2006) and ERS/SKYNET-SKYRAD (Campanelli et al., 2007) inversions. The use of in-situ data from OPC measurements as a benchmark allows for an assessment of the reliability of the different methods (Tsekeri et al., 2017) in the presence of coarse-mode dominated aerosols.

### 2.3.1 AERONET

The AERONET inversion employs measurements of direct and diffuse radiation with sun and sky radiometers to retrieve aerosol optical and microphysical particle properties that are representative of the total atmospheric column (Dubovik et al., 2000, 2006). The AERONET algorithm assumes a vertically homogeneous atmosphere and a mono-component aerosol with a single complex refractive index. AERONET inversion products include the particle size distribution, the complex refractive

index, the scattering phase function, the single-scattering albedo, and spectral and broad-band fluxes. Size distributions obtained from AERONET measurements of mineral dust have shown a dominant mode at around 4 to 5 μm in diameter (Müller et al., 2012; Marenco et al., 2018). The AERONET retrieval forces the particle size distribution to zero at 30 μm in diameter. This constraint may therefore lead to an underestimation of the concentration of large particles by AERONET (Ryder et al., 2019). The reported uncertainties for the AERONET size distribution retrievals in the range from 0.1 to 7.0 μm in radius are given as

10% to 35%, while for larger sizes, uncertainties rise up to 80% to 100% (Dubovik et al., 2000, 2002).

### 2.3.2 ESR/SKYNET

SKYNET is an international research network of users of the PREDE Co. Ltd POM sky radiometer with a growing number of instruments now exceeding 100 units. Currently, SKYNET uses two versions (4.2 and 5) of the inversion algorithm SKYRAD to analyse the radiance measurements of the PREDE POM sky radiometers, although other versions are being developed and

currently tested. In order to benefit the international community of users, a re-organisation of the network structure has been initiated (Nakajima et al., 2020).



Although the International SKYNET Data Center (ISDC) has already started data collection and analysis, different regional sub-networks are well established, and develop new research products and test new methodologies (Nakajima et al., 2020). In Europe, the regional sub-network is called the European SKYNET Radiometers network (ESR). In ESR, versions of SKYRAD
software have been adapted to analyse data from CIMEL sun-sky photometers (Estellés et al., 2012). In this analysis the current SKYRAD version 4.2 is used and the corresponding inversions will be called SKYRAD retrievals.

As for the AERONET inversion, the SKYRAD algorithm estimates the size distribution, phase function and surface albedo of aerosols from measurements of diffuse sky radiance (Campanelli et al., 2007). A notable difference to AERONET is, however, that the SKYRAD retrieval does not prescribe an upper boundary for particle size (Estellés et al., 2012).

### 2.3.3 GARRLiC

The GARRLiC retrieval is a synergistic algorithm that combines quasi-simultaneous passive sky-radiance measurements with active lidar measurements during cloud-free conditions. The required input from sun photometer observations includes the total AOD and radiances at 440, 670, 870, and 1020 nm. Concurrently, lidar measurements of the elastic backscatter signals at 355, 532, and 1064 nm are used as input for GARRLiC (Lopatin et al., 2013). The output of the retrieval provides profiles
of aerosol volume concentration together with a column-mean aerosol volume size distribution, spectral refractive index, and spherical particle fraction. GARRLiC also retrieves aerosol optical properties such as the single-scattering albedo, backscatter and extinction coefficients, and the aerosol lidar ratio.

The lidar input enables GARRLiC to account for variations in aerosol stratification. Due to the wider set of input parameters, the GARRLiC retrieval requires fewer assumptions than other algorithms (Bovchaliuk et al., 2016; Tsekeri et al., 2017). In
case of a bi-modal aerosol distribution, GARRLiC provides the flexibility to use a bi-component aerosol model that may have different refractive indices in the fine and coarse mode. In the presence of mixtures of aerosol types with multiple contributions to the fine and coarse mode (e.g. mixture of marine and dust particles, Tsekeri et al. 2017), the algorithm provides an average estimation similar to the AERONET retrieval. We constrain the investigation in this study to one dust mode because the UCASS observations at Cyprus show a dominance of coarse-mode dust particles throughout the atmospheric column.

## 2.4 HYSPLIT backward trajectories

The Hybrid Single-Particle Lagrangian Integrated Trajectory model (HYSPLIT, Stein et al. 2015; Rolph et al. 2017) run with Global Data Assimilation System (GDAS) meteorological reanalysis fields was used to investigate the origin of the observed air masses over Cyprus. Five-day backward trajectories starting at the locations of the remote-sensing sites were calculated for arrival heights between 1.0 and 7.0 km.



## 3  Results and Discussion

### 3.1  Overview of measurements

The A-LIFE field experiment took place between 3 and 29 April 2017 when the DLR-Falcon research aircraft was deployed at Paphos airport. The period between 20 and 22 April 2017 was dominated by south-western airflow with favourable conditions for the transport of Saharan dust to Cyprus. Persistent periods of clear sky made for ideal conditions for remote-sensing observations. Five UCASS OPCs were launched during an intense dust outbreak that lasted from 20 to 22 April 2017. Figure 1 provides an overview of the temporal evolution of the dust plume over Limassol between 19 and 23 of April 2017 in the form of column-integrated parameters measured with two AERONET sun photometers and height-resolved observations with the Polly$^{XT}$ aerosol lidar. The figure also shows the times and locations of the UCASS launches. Four launches were performed from Paphos Airport (34.71 °N, 32.48 °E), while one UCASS sonde (0134 UTC on 21 April 2017) was launched from the lidar site in Limassol (34.7 °N, 33.0 °E) (Ansmann et al., 2019). Table 1 provides an overview of the UCASS launch times together with the time periods and locations of the remote-sensing and airborne measurements used for comparison. The first airborne mission over this period was performed on 19 April; the leading edge of the dust plume was found to be over Malta and moved eastwards across the Mediterranean in the following days.

The AERONET measurements in Figure 1a show the arrival of the dust plume over Limassol in the morning of 20 April 2017 in the form of an increase in AOD that is accompanied by a decrease in the Ångström exponent. The former refers to an increase in aerosol loading while the latter indicates that large particles are present in the atmosphere. The highest AOD of around 0.5 at 500 nm was observed in the morning of 21 April 2017. The AOD stayed fairly constant at around 0.4 for the rest of the day and decreased slightly to 0.35 on 22 April 2017. During this period, the Ångström exponent stayed very constant at 0.3. The sudden shift in AOD to 0.1 and in Ångström exponent to 0.9 in the morning of 23 April 2017 indicates the departure of the dust event from Limassol and the return to background conditions.

This narrative is corroborated and complemented by the height-resolved lidar observations in Figure 1b. The first faint traces of the dust plume were detected between 4 and 5 km height in the afternoon of 19 April 2017. The main plume arrived at a height of 3 km at 0400 UTC on 20 April 2017. The top of the dust plume reached as high as 7 km at 0600 UTC on 21 April and slowly descended to 4 km until the dust plume departed at 0400 UTC on 23 April 2017. The lidar signal reveals the structure of the dust plume, most notably a thin filament of strong backscatter signal between 2 and 3 km height from 2200 UTC on 20 April 2017 to 1200 UTC on 21 April 2017. The lidar plot shows a homogeneous dust layer in the upper part of the plume and features that correspond to the settling of dust particles over time, i.e. structures that appear at lower heights as the dust plume passed over the lidar station.

The trajectories of the five UCASS launches up to 10 km height are shown in the left panel of Figure 2. It generally takes about 40 minutes for the balloon to reach such an altitude, though it depends on the individual ascent rate. The first and second UCASS launched from Paphos and Limassol, respectively, headed eastwards. The first unit reached closest to the lidar site which make this case ideal for a comparison of UCASS measurements with the findings of the remote-sensing retrievals that include the lidar data. UCASS units from later launches headed to the north east and show that the main wind direction changed





during the passage of the dust plume. The right panel in Figure 2 shows the tracks of the DLR-Falcon aircraft during research
240 flights on 20 and 21 April 2017. The close proximity makes these cases most suitable for a comparison of the measurements
during the first and third UCASS launch to those of airborne in-situ instruments. Details on the distance between the respective
observations are provided below.

Figure 3 shows 120-h backward trajectories of air masses arriving at 2, 3, 5 and 7 km height over Limassol at 0100 UTC on
21 April 2017, i.e. at the time of the second UCASS launch. Trajectories are shown for this UCASS launch as it coincides with
the presence of the unusual filament structure over Limassol in Figure 1b. The trajectories follow similar pathways for the other
launches. They reveal that these air parcels were lifted from dust source regions in North Africa, crossed the Mediterranean,
and reached Cyprus within three days. The air parcels arriving at 2 and 3 km height originated from northern Libya while those
arriving at 5 and 7 km height originated from Algeria, Morocco, and Mauritania. The difference in source region and transport
time for air arriving at different altitudes might lead to differences in the observed particle size distributions at those heights
(Weinzierl et al., 2009, 2011; Ryder et al., 2013, 2018). The inspection of dust composites derived from measurements with
the Spinning Enhanced Visible and Infra-red Imager (SEVIRI) on the Meteosat Second Generation satellite (Schepanski et al.
2007, not shown) shows that dust was mobilised in the northern part of Cyrenaica (i.e. north eastern Libya) about 24 h before
the observations of the second UCASS launch and transported directly to Cyprus.

### 3.2 Number concentration profiles

Figure 4 shows the particle number concentration from the first and third UCASS launch and the DLR-Falcon measurements
together with the distance between the locations of the respective measurements. The distance between the observations is
below 40 km up to a height of 2.4 km and around 80 km above. The first UCASS launch in the evening of 20 April 2017 from
Paphos shows the highest particle concentrations of more than $30\,cm^{-3}$ between 4.5 and 5.5 km height. The meteorological
profiles (not shown) reveal temperature inversions at the bottom and top of the dust layer. They also show a dry lower and a
much more humid upper part of the dust plume with 30% RH between 1 and 3 km and 80% RH between 3 and 5 km, respec-
tively. Despite the rather large spatial distance of the observations, there is close resemblance of the number concentrations
measured with the UCASS and aboard the aircraft in the lower and upper part of the dust plume with an average ratio of
0.92. The discrepancy increases as the horizontal distance between the observations increases, i.e. beyond 70 km above 2.5 km
height. The discrepancy could be attributed to the large amount of smaller particles, as the lowest size bin for the UCASS
number concentration is $0.6\,\mu m$, whereas for the CAPS is $0.79\,\mu m$. Nevertheless, the close resemblance of both profiles is
indicative of the spatial and temporal homogeneity of the dust plume and suggests that differences in time and location of the
observations do not necessarily inhibit a comparison of the measurements.

Figure 4b shows the UCASS measurements during the third launch on 21 April 2017 from Paphos. The highest particle
concentrations are still found in the upper part of the dust plume which now extends from about 3 to 6 km. While the number
concentration exceeds $20\,cm^{-3}$ in this layer, the higher concentrations above $30\,cm^{-3}$ as measured during the first launch were
now found only at around 4 km and just below 5 km height. The descent of the larger number concentrations to lower heights
is indicative of gravitational settling of particles in the uppermost part of the dust plume during transport. The humidity profile





(not shown) reveals drier air throughout the extend of the dust plume compared to the first launch with RH spanning from 20% to 65%. The aircraft observations during that day (Table 1 and Figure 2) took place in close proximity to the UCASS track with a horizontal distance of less than 15 km below 3 km height and 20 to 50 km in the upper layer of higher particle concentrations. Consequently, the number concentrations of the measurements with the UCASS and aboard the DLR-Falcon agree even on the fine structures of the dust plume during that time. Below 3.2 km height, observations with UCASS and aboard the aircraft show number concentrations in the range from 5 to 10 cm$^{-3}$ and both profiles resolve a layer of increased particle concentration at around 2.0 km height. In the upper part of the dust plume above 3.2 km height, the lower and upper boundaries of the layer with number concentrations above 10 cm$^{-3}$ at 3.4 and 6.4 km, respectively, are resolved by UCASS and airborne measurements within 200 m height, despite the increasing in the spatial distance of the observations to 60 km. Within this layer, the UCASS and the airborne instruments detect peak concentrations of 35 cm$^{-3}$ at 4.5 km height and in the range from 5.0 to 5.4 km height. Both also resolve the decrease to number concentrations of around 20 cm$^{-3}$ at 4.8 km height.

Overall, the particle number concentration and size distributions observed with UCASS over Cyprus are similar in magnitude and shape, respectively, to what has been reported from aircraft measurements in previous studies. Observations during the ICE-D and AER-D (Ryder et al., 2018; Liu et al., 2018) over the west African coast showed particle number concentrations of up to 45 cm$^{-3}$ in the size range between 0.5 and 20.0 μm in diameter. Particle number concentrations within the dust layers observed during SAMUM in Morocco (Weinzierl et al., 2009, 2011) and SALTRACE over the tropical Atlantic (Weinzierl et al., 2017) decrease from nearly 1000 to 0.001 cm$^{-3}$ in the diameter range from about 0.1 to nearly 40.0 μm. In-situ observations of central Saharan dust size distributions during Fennec (Marsham et al., 2013; Ryder et al., 2013) with wing-mounted instruments that measured particle diameters between 0.1 and 100 μm gave number concentrations of up to 1000 cm$^{-3}$ (Ryder et al., 2018). Looking at the size distribution of SALTRACE, the highest particle number concentrations are found between 0.5 and 1 μm. The particle number concentration decreases towards the larger sizes. A closer look at the size range between 0.4 and 20.0 μm, which is closer to the measurement capability of UCASS, reveals number concentrations spanning from 0.01 to 100 cm$^{-3}$. Note that most of the observations listed above have been conducted much closer to dust sources compared to the measurements at Cyprus presented here. Hence, it can be concluded that the UCASS observations give values that are in line with data from airborne campaigns.

### 3.3 Layer-averaged number size distributions

A closer look at layer-mean particle size distributions from the measurements with the UCASS and the research aircraft on 20 and 21 April 2017 is provided in Figure 5. The extent of the considered height layers is marked in Figure 4. The particle number concentrations are similar for both observation days, particularly at sizes below 5 μm, with maximum values between 10 and 50 cm$^{-3}$. However, the two largest UCASS size bins with bin centres at 8.4 and 12.1 μm tend to detect fewer particles than the instruments aboard the research aircraft, though the numbers of around 0.1 cm$^{-3}$ are well within the respective error bars. The size distributions resemble each other very well in terms of their shape at all height layers on 20 April 2017. The particle number concentrations agree within their error bars for most size bins. Only the concentrations of UCASS bin four for the layers from 3.0 to 4.0 km and 4.0 to 5.0 km height are smaller than the ones observed by the airborne instruments and





cannot reach the latter within their measurement error. The observations on 21 April 2017 also resemble each other very well in terms of the shape of the size distribution at the lowermost (1.0 to 3.0 km height) and uppermost (5.1 to 5.7 km height) layer on 21 April 2017. Pronounced discrepancies in the observations without an overlap of the error bar is found in the height layer between 3.6 and 4.6 km. Figure 5e shows that there is a consistent difference in the particle number concentration of about $10\,\mathrm{cm}^{-3}$ for the measurements of the two instruments within that layer. This is probably due to a spatial mismatch of the layer at the two locations as indicated by the similar structure of the profiles. Overall, the aircraft and UCASS observations give very similar magnitudes and shapes of the size distributions in the uppermost two layers on both 20 and 21 April 2017.

A mean effective diameter of $2.4 \pm 0.3\,\mu\mathrm{m}$ was found from the UCASS measurements in the size range from 0.4 to 14 µm. Observations during AER-D and ICE-D at Cape Verde gave a mean effective diameter of 4.0 µm (Ryder et al., 2018) and 5.0 to 6.0 µm (Liu et al., 2018) for the size ranges from 0.1 to 100.0 µm and from 1.0 to 40.0 µm, respectively. Measurements during SAMUM (Weinzierl et al., 2009) gave effective diameters of about 6.5 µm for measurements that covered particle sizes up to 100 µm in diameter. The values obtained from our UCASS measurements are slightly lower than those reported in the literature. This is likely due to the fact that the UCASS as deployed during A-LIFE measured only up to particle diameters of 14 µm while its nominal detection range extends to 17 µm. In any case, a comparison of the effective diameters from different measurements may not be comprehensive due to the different source regions and travelled distances of the observed dust particles from the different observations.

### 3.4 Columnar size distributions

A comparison of the columnar aerosol volume size distribution from the GARRLiC, AERONET, and SKYRAD retrievals and the first, and third UCASS launches is presented in Figure 6. Figure 6a also includes the airborne in-situ observations with the Falcon aircraft. All distributions in Figures 6a and b show a predominance of coarse-mode particles with comparable volume concentrations of the different coarse-mode peaks. During Launch 1, the UCASS and GARRLiC retrieval show a different shape of the volume size distribution compared to the ones retrieved by the sun photometer inversions. Both AERONET and SKYRAD show a single coarse mode that peaks between 3 and 5 µm, while GARRLiC and UCASS give a coarse mode with two peaks. AERONET observed the highest concentration of $0.12\,\mu\mathrm{m}^3/\mu\mathrm{m}^2$ between 3.4 and 4.5 µm diameter, whereas SKYRAD's size distribution peaks at 3.4 µm with a concentration of $0.13\,\mu\mathrm{m}^3/\mu\mathrm{m}^2$. The UCASS observed its highest concentration of about $0.1\,\mu\mathrm{m}^3/\mu\mathrm{m}^2$ at 5.5 µm in diameter and a second mode at 2.8 µm. The two coarse modes retrieved by GARRLiC are at 2.0 and 7.7 µm in diameter. It is noteworthy to mention that the first UCASS unit was launched about 2 hours 40 min after the considered sun-photometer measurement as outlined in Table 1. The first UCASS launch shown in Figure 6a is also the only case for which a column-integrated volume size distribution is available from the observations aboard the research aircraft. These independent airborne in-situ measurements also find a bi-modal coarse mode which supports the results of the UCASS measurements and suggests that the findings of the GARRLiC retrieval are closer to reality than those from AERONET and SKYRAD.

Post-processing was applied to the UCASS data. In addition, further laboratory measurements with a set-up comparable to the conditions encountered during the launches on Cyprus were performed to examine if the observed bi-modal size distribu-



tions in Figure 6 could be the result of an instrumental artefact. Mono-modal sample materials were used in these laboratory tests (Smith et al., 2019). The corresponding UCASS measurements also showed only mono-modal size distributions. This lead us to reject the idea of a systematic instrumental error. Hence, the bi-modal coarse mode might be a special characteristic of the origin of the observed air masses. The observed bi-modal peak may be caused by the following reasons: (i) the diversity

of sources across the African basin whose mineralogy can lead to intrinsic differences in the properties of the emitted particles (e.g. size distributions, chemical composition) (Engelstaedter et al., 2006; Coz et al., 2009), (ii) cloud processing during transport which could cause aggregation of particles that were collected by droplets that evaporated at a later stage or wash-out of larger particles (Matsuki et al., 2010), (iii) gravitational settling of particles for longer transport times compared to freshly emitted dust after about one day of transport might lead to the systematic removal of large particles, particularly in the upper

part of dust plumes (Ellis and Merrill, 1995; Maring et al., 2003), or (iv) dust electrification that could counteract gravitational settling by creating an electric field within the dust layer (Nicoll, 2012). A similar bi-modal coarse-size distribution was also observed during the Puerto Rico Dust Experiment (PRIDE, Reid et al. 2003) and Fennec SAL (Song et al., 2018). However, neither study provides further discussion of these observations.

The second UCASS launch on 21 April 2017 was performed about 3.0 h and 1.5 h before the first sun photometer and lidar

measurements, respectively (Table 1). This is likely to have an effect on the UCASS comparison in Figure 6b as the aerosol conditions varied strongly during that period (Figure 1). The UCASS size distribution peaks at 5.8 μm with a concentration of 0.13 μm³/μm². This peak is also resolved by the GARRLiC-derived size distribution, though it is located at 2.6 and 5.9 μm with concentrations of 0.24 and 0.18 μm³/μm², respectively. The AERONET-derived size distribution shows a peak concentration of 0.22 μm³/μm² between 3.4 and 4.5 μm particle diameter. The SKYRAD retrieval gives a peak concentration of

0.21 μm³/μm² at a coarse-mode diameter of 3.4 μm which is the smallest compared to the other observations. Although the sun photometer inversions rely on the same input data sets, it is found that the SKYRAD size distribution is shifted to smaller sizes compared to AERONET. This is surprising as SKYRAD does not force the size distribution to zero at larger particle diameters (Campanelli et al., 2007) and, in principle, would enable the retrieval of size distributions with larger coarse-mode diameters than AERONET. This particular property of the AERONET retrieval is likely to produce the artificial fine-mode

peak at around 0.15 μm that is absent in the SKYRAD size-distributions (Dubovik et al., 2006).

The peaks of volume size distribution from the sun photometer inversions are found to be systematically at smaller particle sizes than the observations from GARRLiC and the UCASS. A similar shift towards larger particle size was also observed from in-situ measurements aboard the Falcon compared to AERONET-derived size distributions during SAMUM (Müller et al., 2012). A similar tendency between AERONET size distributions and in-situ measurements was observed during DABEX in

the Sahelian west Africa basin (Osborne et al., 2008). During the SAVEX-D/AER-D campaign at Cape Verde, the AERONET retrievals also showed a similar single coarse mode shifted towards a smaller radius compared to in-situ measurements from aircraft (Estelles et al., 2018). Simultaneous retrievals from SKYRAD (performed on Prede POM radiometers) also determined a slight shift of the coarse mode to a smaller radius, although in this case the coarse mode was broader or even bimodal, depending on the SKYRAD version used (Nakajima et al., 2020). The coarse mode retrieved by GARRLiC shows a consistent shift

towards larger sizes when compared to the AERONET output (Benavent-Oltra et al., 2017; Lopatin et al., 2013; Bovchaliuk





et al., 2016). This feature is generally attributed to the additional information from the backscatter lidar profiles that provides GARRLiC with extra information on the particle size. In addition, the restriction of the AERONET (and hence also the GAR-RLiC) data inversion scheme to a particle diameter smaller than 30 $\mu$m may lead to an underestimation of the concentration of coarse-mode particles (Müller et al., 2012).

A closer look at the UCASS measurements during the second launch is provided in Figure 7 in terms of the profile of total number concentration and volume size distributions averaged over four height layers. Figure 7c shows that the altitude range between 2.8 and 3.1 km is dominated by particles with a mode diameter of 8.4 $\mu$m. In contrast, all other layers show an up to an order of magnitude smaller concentration of particles with such large diameters. Hence, the thin filament of dust particles observed in the morning of 21 April 2017 is the major contributor to the coarse-mode in the columnar size

distribution in Figure 6b. This structure was confined to a very small height range and only lasted for about 12 h. It had already disappeared from the ground-based remote-sensing sites during the time of the DLR-Falcon research flight that day. Longer transport times translate to a longer time period during which large particles are exposed to gravitational settling. This effect is most pronounced at higher altitudes where no particles can settle into the layer from above. Figure 3 indicates that the aerosols observed at 5 and 7 km height have been transported over longer distances than those at lower altitudes. As stated before, MSG-

SEVIRI imagery shows active dust sources in north eastern Libya about 24 h before the observations at Limassol. Backward trajectories corroborate that dust emitted from these sources would was transported directly to Cyprus. It is likely that this is the origin of the thin filament observed in the morning of 21 April 2017. It is noteworthy to state that the first sun photometer observations (used for AERONET, SKYRAD, and GARRLiC retrievals) took place after sunrise (0427 UTC), when the dense aerosol filament over Limassol had changed its appearance and extended in depth.

The overarching message of Figures 6 and 7 is twofold. Firstly, reasonable agreement can be found between the UCASS measurements and data from remote-sensing observations in case of homogeneous dust properties and optimum temporal matching of the observations (Figure 6a). Under such conditions, the more complex GARRLiC retrieval which is based on a larger set of input data is capable of better resolving the features of the UCASS in-situ measurements, i.e. the double-peak in the coarse mode. Secondly, the requirement for homogeneous aerosol conditions is vital, if observations at different

times are compared or used as combined input to a retrieval. In that context, Figure 6b provides some insight into the actual spread of findings that can result from extreme variations in the aerosol situation such as changes in total aerosol load or the vertical distribution of the particles. This is particularly important when using passive remote-sensing data for the validation of vertically-resolved measurements as they provide no information on aerosol stratification.

### 3.5    Extinction coefficient profiles

Figure 8 shows the aerosol extinction coefficient profiles as calculated using the UCASS observations during the four launches listed in Table 1. Lidar profiles of the extinction coefficient measured by the Polly[XT] at Limassol were derived using two methods. Firstly, the extinction coefficient was obtained without assumptions using the Raman method (Ansmann and Müller, 2005). Secondly, the likely range of extinction coefficients was estimated by multiplying the particle backscatter coefficient obtained using Klett's method with the lower and upper limits of reasonable dust lidar ratios for Cyprus of 40 and 60 sr,



respectively (Nisantzi et al., 2015). Generally, the profiles of the extinction coefficients from the UCASS and lidar at the lowermost layers are in a reasonable agreement with values below $100\,\mathrm{Mm}^{-1}$. Discrepancies are more pronounced for the observations within the elevated layers.

The extinction coefficient profiles were integrated with height to obtain an estimate of AOT that can be compared to the sun photometer measurements. This comparison is shown in Figure 1a and in Table 2. The lower lidar-estimate of AOT derived

using Klett's method and a lidar ratio of $40\,\mathrm{sr}$ shows the best agreement with the independent sun photometer observations at both Paphos and Limassol with differences (lidar minus AERONET) ranging from -0.07 to 0.03 for absolute AOT values between 0.35 and 0.52. The UCASS-derived AOTs show no consistent behaviour when compared to the lidar and sun photometer values. The value of 0.39 obtained from Launch 5 agrees best with both lidar (0.36, LR=$40\,\mathrm{sr}$) and sun photometer (0.43 at Limassol and 0.44 at Paphos). Launch 2 with an AOT of 0.65 is at the upper end of the lidar estimate which is 0.63 when

using Klett's method with a lidar ratio of $60\,\mathrm{sr}$. However, Launch 1 and Launch 3 give AOTs of 0.65 and 0.79, respectively, that are well above the remote-sensing estimates that range from 0.32 to 0.58. Figure 8 confirms that this is due to the elevated layers characterised by peak particle concentrations. Particularly, this occurs when UCASS-derived extinction coefficients are as high as $300\,\mathrm{Mm}^{-1}$. Much lower extinction coefficients of 70 to $150\,\mathrm{Mm}^{-1}$ are found from the different analyses of the lidar measurements (Klett, Raman). As these layers are characterised by an increased concentration of larger particles, there

is reason to believe that the current UCASS extinction conversion is not universally applicable to different aerosol conditions. The refractive index and the size-dependent extinction efficiency are the main factors in the retrieval of extinction coefficients from the UCASS measurements. The large particles in the elevated layers might therefore be of different chemical composition compared to those at lower layers. This is supported by the backward trajectories in Figure 3 which indicate different source regions for air arriving at different height levels. Alternatively, the extinction efficiency used in the current conversion

might be representative only for situations dominated by smaller particles for which the effect of particle non-sphericity is less pronounced. Comparisons with extinction coefficient, which is a secondary-order parameter derived from UCASS data, are therefore questionable and require further investigation that is beyond the scope of this study.

## 4  Summary and conclusions

We have presented findings from balloon-borne UCASS optical particle counter measurements of mineral dust conducted

over Cyprus in April 2017 during the A-LIFE experiment. The UCASS launches were embedded in research activities that included airborne in-situ measurements with the DLR-Falcon research aircraft as well as ground-based remote sensing with advanced aerosol lidars and sun photometers. This setup allows for a comprehensive evaluation of the quality of the UCASS measurements as well as an assessment of a variety of remote-sensing retrievals.

The highest particle number concentration observed by the UCASS was found during the first launch, with values of up to

$50\,\mathrm{cm}^{-3}$ within a layer from 3 to 5 km height. Aircraft observations gave slightly lower values with a maximum of $40\,\mathrm{cm}^{-3}$. The UCASS profile of number concentration during the third launch resembles the aircraft observations in the vertical structure as well as the dust load.





During the first launch, layer-averaged particle size distribution from the UCASS measurements resembles those measured by the instruments aboard the aircraft in the size range between 2 to 5 μm. Larger differences in the particle number concen-

tration are found for particle sizes larger than 5 μm, though these are still within the error bar of the measurements. During the second launch conducted from Limassol, a thin filament of dust was observed between 2.8 and 3.1 km height. This feature revealed a dominance of very large particles with average mode diameter of 8.4 μm. In contrast, the layers below 2.2 km and above 4.8 km height were dominated by lower concentration of coarse-mode particles. Furthermore, it was found that the number concentration of large particles decreased with altitude. This is likely to be a result of gravitational settling.

Column-integrated particle volume size distribution were calculated from the UCASS measurements for a comparison to the findings of remote-sensing retrievals. For the first launch, results from the GARRLiC retrieval are the only ones that reproduce the bi-modal coarse mode detected by UCASS while the AERONET and SKYRAD inversions give a single coarse mode. Nevertheless, the AERONET-derived size distribution particle concentrations are similar to the UCASS measurements in the size range from 1 to 6 μm. In contrast, the SKYRAD-derived size distribution shows a coarse-mode peak at 2.0 to 3.0 μm which

is in line with the first coarse-mode peak of the GARRLiC-derived size distribution but smaller then the other inferred coarse-mode peak diameters. During the second launch, the volume size distributions obtained by the UCASS and GARRLiC peak at a particle diameter of around 6.0 μm. However, GARRLiC also gives a second pronounced peak at around 3.0 μm that is hardly visible in the UCASS measurement. In addition, large discrepancies on the shape and maximum of the volume size distribution were observed between the UCASS and the retrievals obtained by sun photometer data alone, i.e. the AERONET and SKYRAD

inversions. This is attributed to the temporal difference between the observations of 3 h and the strongly heterogeneous dust layering: the sun photometer observations were performed when the thin dust filament observed by the UCASS and lidar had dissipated into the layer below the filament. . Overall, UCASS measurements of particle concentrations and size distributions are found to be reasonably in line with coincident observations with research aircraft and remote-sensing instruments. The low-cost and disposable nature of the instrument therefore makes it a attractive tool for the in-situ profiling of atmospheric

particle concentrations in the framework of field experiments and long-term observations.

The comparison with secondary-order parameters, i.e. extinction coefficient, is yet questionable and requires further investigation beyond the scope of this study. Given the relatively low cost of the disposable UCASS, it provides a promising new opportunity for in-situ measurements of particle size distributions within different aerosol layers and for validation studies between remote-sensing and in-situ observations (Sawamura et al., 2017), not only for optical data but also microphysical data.

*Data availability.*  The measurement data used in this study are available from the authors upon request.

*Author contributions.*  MK and MT performed the UCASS launches from Cyprus. MK and HS assembled and calibrated the UCASS units used during A-LIFE and analysed the measurements. ZU and HS developed the concept of UCASS and the associated data analysis techniques. HB analysed the PollyXT data. VE performed the SKYRAD inversions. MD and BW performed and analysed the airborne measure-



ments. MK and AT performed the GARRLiC inversions. MK and MT prepared the initial version of the manuscript. All authors contributed
to the discussion of the findings and the revision of the draft paper.

*Competing interests.* The authors declare that they have no conflict of interest.

*Acknowledgements.* The authors gratefully acknowledge the NOAA Air Resources Laboratory (ARL) for the provision of the HYSPLIT
transport and dispersion model and/or READY website (http://www.ready.noaa.gov, last access on 10.12.2019) used in this publication.
AERONET data for the stations Paphos and Limassol have been obtained through the AERONET portal (http://aeronet.gsfc.nasa.gov, last
access on 28.11.2019). We thank the PIs for maintaining the two sites. This research has received funding from the Royal Society (Royal
Society Research Grant RG160071), the European Research Council (640458, A-LIFE), the European Union's Horizon 2020 research and
innovation program (654109, ACTRIS-2), and the Franco-German Fellowship Programme on Climate, Energy, and Earth System Research
(Make Our Planet Great Again – German Research Initiative, MOPGA-GRI 57429422) of the German Academic Exchange Service (DAAD),
funded by the German Ministry of Education and Research. V. Estellés acknowledges the support of the Spanish Ministry of Economy and
Competitiveness (MINECO) and the European Regional Development Fund (FEDER) for the ESR/SKYNET activities, through Project
CGL2017-86966R.



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



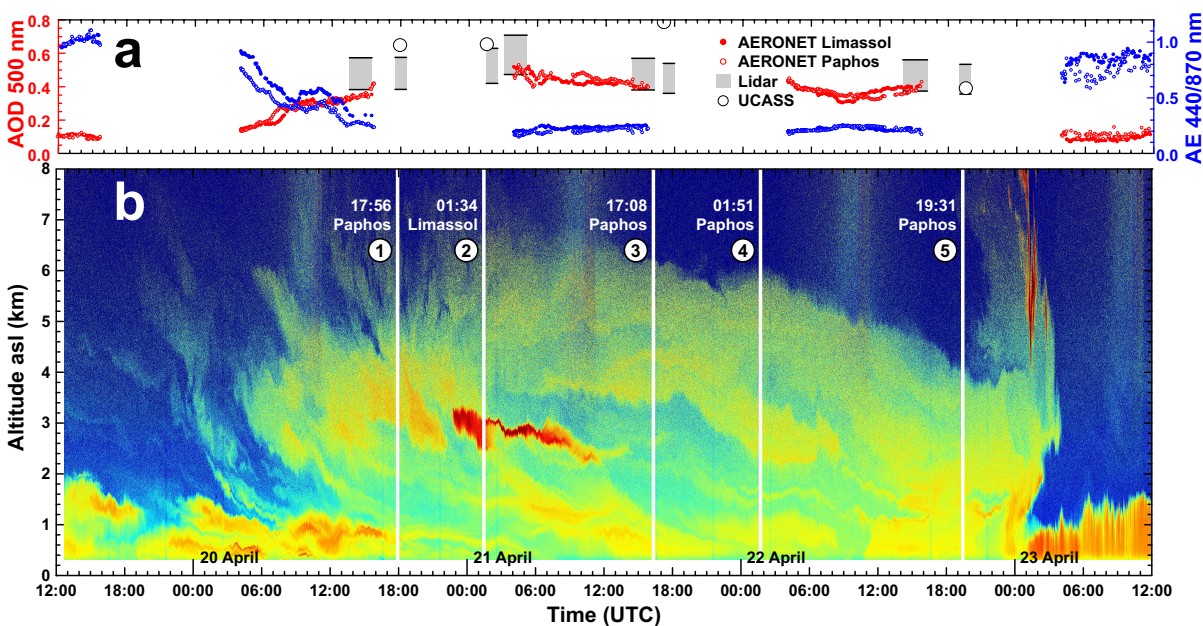

**Figure 1.** Overview of the aerosol conditions at Limassol (34.7 °N, 33.0 °E) during the 4-day period from 20 to 23 April 2017 in terms of (a) the aerosol optical depth (AOD) at 500 nm (red) and the Ångström exponent (AE) for the wavelength pair 440/870 nm (blue) as obtained from AERONET sun photometer observations at Limassol (filled dots) and Paphos (open dots), UCASS (532 nm, black circles), and Polly lidar measurements (grey bar spanning the range of values obtained from multiplying the 532-nm backscatter coefficient with 40 sr and 60 sr, respectively); and (b) the range-corrected signal at 1064 nm as measured with the Polly lidar. Lines and numbers mark the times and locations of UCASS launches. Low backscatter signal (low aerosol concentrations) is shown in blue while very high backscatter signal (dense aerosol layers) is shown in red.





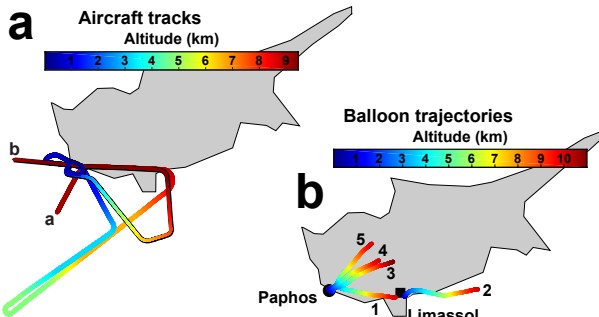

**Figure 2.** Flight tracks of (a) the DLR-Falcon aircraft and (b) the UCASS sondes launched from Paphos and Limassol. The Falcon tracks marked a (black border) and b (no border) refer to the research flights on 20 and 21 April 2017, respectively.

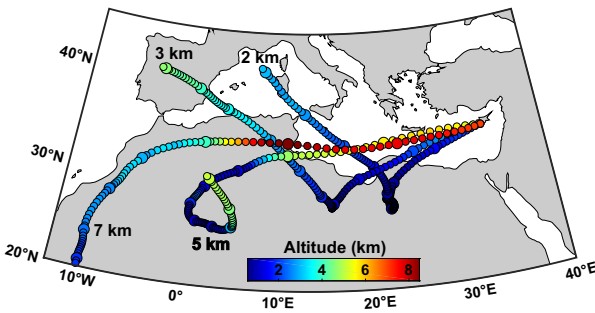

**Figure 3.** 120-h HYSPLIT backward trajectories starting over Limassol at 0100 UTC on 21 April 2017. Colour coding refers to the height of the trajectories. Intervals of 12 h are marked by increased circle size. The numbers in the plot refer to the arrival height over Limassol.





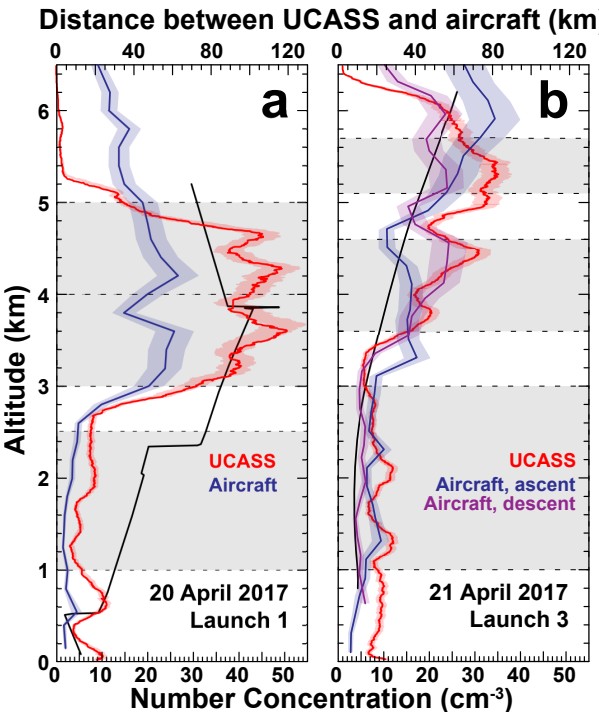

**Figure 4.** Total number of particles in the diameter range from 0.6 μm to 13.9 μm counted by UCASS (red) and with the CAPS instrument aboard the Falcon aircraft (blue and purple, size range from 0.79 to 14.0 μm, shading marks the standard deviation) measured (a) during the first UCASS launch on 20 April 2017 and (b) during the third UCASS launch on 21 April 2017. Red shaded areas refer to the effect of a counting uncertainty of 8% (y error) as stated in (Smith et al., 2019). The black lines mark the horizontal distance between the location of the observations from the UCASS and aircraft. Gray areas refer to the height layers considered in Figure 5.





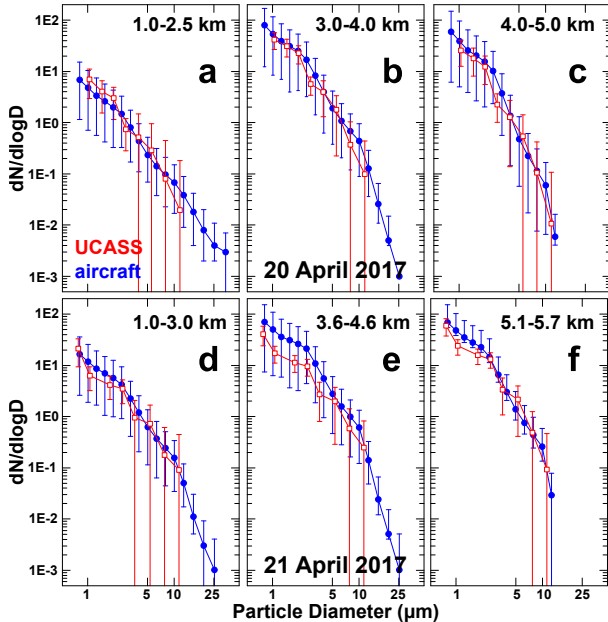

**Figure 5.** Particle number size distribution observed by UCASS (red) and with the CAPS instrument aboard the Falcon aircraft (blue) on 20 (a-c, launch 1) and 21 April 2017 (d-f, launch 3) for the height layers marked in Figure 4. Error bars refer to the standard deviation of the measurements.





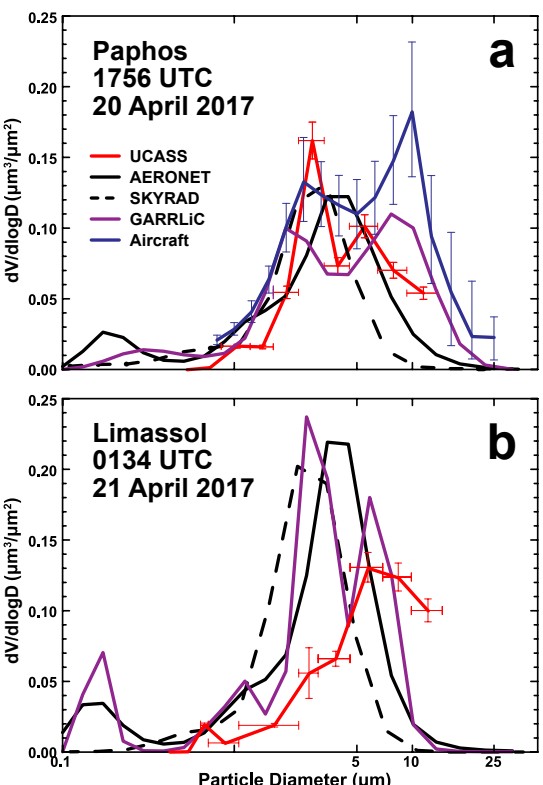

**Figure 6.** Total-column volume size distribution from the UCASS (red), AERONET (black solid), SKYRAD (black dashed), GARRLiC (purple), and aircraft (blue, only in a) obtained during (a) the first launch on 20 April 2017 and (b) the second launch on 21 April 2017. For clarity, only error bars of the UCASS and CAPS measurements are shown. Details on the locations and measurement times are provided in Table 1.





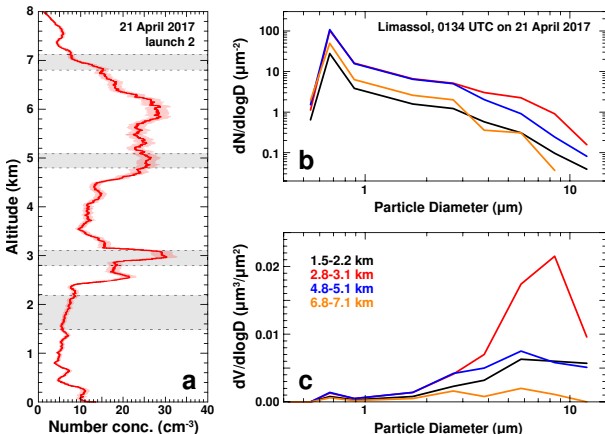

**Figure 7.** UCASS measurements during the second launch at 0134 UTC on 21 April 2017 from Limassol: (a) height profile of the total particle number concentration as in Figure 4 and (b) particle number and (c) volume size distributions averaged over four selected height levels as indicated in (a).

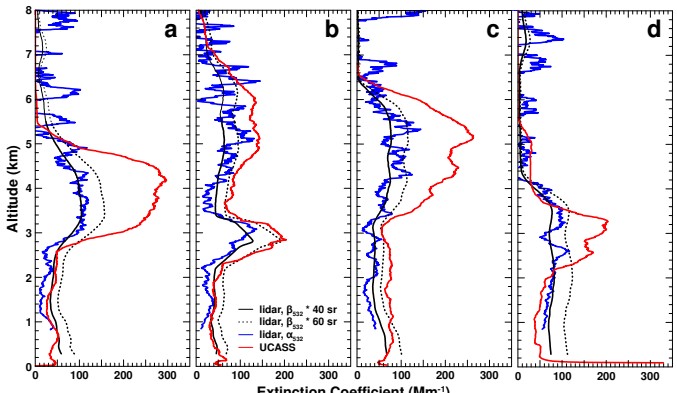

**Figure 8.** Aerosol extinction coefficient profiles from UCASS (red) and the Raman lidar PollyXT at Limassol. Lidar profiles refer to extinction coefficients obtained using the Raman method (blue) or by multiplying the aerosol backscatter coefficient derived without the use of Raman signals with lidar ratios of 40 sr (solid black line) and 60 sr (dotted black line).





**Table 1.** Dates and times (UTC) of UCASS launches, aircraft profiles (together with location of observation), and measurements with lidar and sun photometer (SPM) used in this study. UCASS units were launched from Paphos except for the launch at 0134 UTC on 21 April 2017 which was performed next to the lidar site at Limassol. Lidar 1 refers to the time period used for the comparison of AOD and extinction coefficients in Figures 1 and 8, respectively. Lidar 2 marks the time period used for the combined lidar-SPM retrievals with GARRLiC in Figure 6.

|  | UCASS | Lidar 1 | Aircraft | | Lidar 2 | SPM |
|---|---|---|---|---|---|---|
| 20 April 2017 | 1756 | 1830-1930 | 1738-1833 | W of Cyprus | 1330-1530 | 1511 |
| 21 April 2017 | 0134 | 0130-0230 | 1148-1248 | W of Cyprus | 0305-0505 | 0427 |
| 21 April 2017 | 1708 | 1700-1800 | 1408-1507 | SW of Cyprus | 1415-1615 | 1449 |
| 22 April 2017 | | | UCASS launch unsuccessful | | | |
| 22 April 2017 | 1931 | 1900-2000 | 0815-0915 | S of Cyprus | 1410-1610 | 1512 |

**Table 2.** Column AOD derived from the integration of the extinction coefficient profiles in Figure 8 for the times of the UCASS launches (see Table 1). AODs are also shown in Figure 1a. UCASS and lidar AODs are at 532 nm. AERONET AOD is at 500 nm. Lidar Raman refers to the Raman solution of the lidar equation while the other two lidar values refer to an analysis following Klett's method with lidar ratios of 40 and 60 sr, respectively.

| | UCASS launch number | | | |
|---|---|---|---|---|
| | 1 | 2 | 3 | 5 |
| UCASS | 0.65 | 0.65 | 0.79 | 0.39 |
| Lidar Raman | 0.32 | 0.42 | 0.32 | 0.31 |
| Lidar, S=40 sr | 0.38 | 0.42 | 0.36 | 0.36 |
| Lidar, S=60 sr | 0.58 | 0.63 | 0.56 | 0.53 |
| AERONET Limassol | 0.35 | 0.52 | 0.39 | 0.43 |
| AERONET Paphos | 0.42 | 0.46 | 0.40 | 0.44 |