# Peer review of "Measurement report: Balloon-borne in-situ profiling of Saharan dust over Cyprus with the UCASS optical particle counter"

_Atmospheric Chemistry and Physics, 2020_

## Referee Comment (RC1) · Anonymous Referee #1 · 9 Dec 2020

This paper aims to validate a state-of-the-art sonde for dust profiling based on an in-situ optical particle counter approach. The experimental design is really interesting because it can help, for instance, to explore the lowermost troposphere where lidars are typically blind, leading to complement their measurements for modelling applications. The manuscript is very well structured, allowing for a clear comprehension of the research involved. However, I found some issues to be addressed, mainly minor comments.

General comments:

lines 107-108: It is supposed that the use of Mie scattering has a small effect on the

calculated size distributions even in the presence of non-spherical particles. Can you provide a quantification of this?

lines 188-189: In the instrumentation section (2.2.) is mentioned that the PollyXT allows for measuring polarized components at 355 and 532 nm. It seems (from the sentence in lines 188-189) that only total signals at these wavelengths are used. Does not GARRLIC use polarized components?

lines 193-199: it would be nice to provide an estimation of the uncertainties of the GARRLIC derived products.

Lines 226-233: How are you sure what is dust and what is not? Taking into account the different origin of air masses shown in figure 3, I recommend to include in fig 1 an additional panel ploting information from depolarization (at least volume linear depol ratio, but particle depol ratio is preferred). This also brings me an additional point. What is the uncertainty for depolarization products? How were those channels calibrated? A detailed description is not needed, some relevant references should be enough.

Lines 413-414: How did you deal with the incomplete overlap region to estimate AOTs?

LInes 426-428: Because you used GRASP in this study combining Sun-photometer and PollyXT data, profiles of refractive index and single scattering albedo can be retrieved. This information might support your argument here.

Figure 1: For the sake of clearness, I recommend to use symbols or colors with more contrast for Limassol and Paphos data.

Figure 4: Is there no information about the horizontal distance above roughly 5 km for the case launch 1?

Figure 8: Why do the Klett profiles shown here? It is well-known that this method can not provide consistent extinction information. Even more, why is the full overlap height different from the extinction and the backscatter profiles shown here?

[Figure]

---

## Referee Comment (RC2) · Anonymous Referee #2 · 7 Jan 2021

General comments:

This paper presents new results from a balloon-borne sensor, UCASS, demonstrated and evaluated in dust aerosol during a field campaign in Cyprus. The results demonstrate the usefulness of the instrument and its ability to explore the nature of dust aerosols in the atmosphere. The paper performs a comparison of the instrumental UCASS data against in-situ aircraft observations and ground-based remote sensing. This is a challenging task and the authors present valuable conclusions with regard to the retrieved size distributions, the importance of vertically-resolved information and the ability of UCASS to provide extinction coefficients. The paper is well-written and

well presented, and I recommend publication following the minor corrections listed be-
low.

——

Title: I suggest removing 'Measurement Report' or rewording it in to the rest of the title
– having it at the start of the sentence with a colon implies that it is a particular type of
article/publication at ACP, which is not the case.

Abstract

L1-2 – is this the first publication of UCASS measurements? What is novel about the
new measurements/data collected? This should be clearly communicated here or later
in the abstract.

L13-14 – "An overestimation of the extinction coefficient of a factor of two was found for
layers with particle number concentrations that exceed 25 cm$-3$." An overestimation
of lidar or UCASS? Please provide an indication/quantification of the fraction of lay-
ers/instances where this disagreement is found, so the reader can put the differences
into context.

An additional sentence should be given at the end of the abstract to sum up the overall
abilities of UCASS and it's potential to provide future measurements and/or insights
into environmental data.

Introduction

Paragraph 1 – Stocker et al. (2013) is missing from the reference list. Citing Stocker
twice for this broad paragraph should be avoided – many other papers cover these
topics and could be cited here.

L37 – "The main zones of cyclogenesis. . ." – this sentence is unclear.

L61 – "with the exception of wing-mounted probes" -> "with the exception of airborne
wing-mounted probes"

L81-82 – are these diameters optical diameters?

L100 – please give the particle concentration in cm-3 as this is more interperable and in line with units used in the abstract.

L101 – Equation $n\_i=C\_i/V$ – should V be lower case as defined in the previous sentences?

Section 2.1 – is it possible to retrieve data from the UCASS descent as well as the ascent?

L114 – please confirm that effective diameter is calculated over the full measured size range (several previous observational studies have selected only part of the size range).

Equation 4 – why is i summed up to the value of 10?

L145 – is this a CIP15?

L148 – insert 'optical diameter' for the CAS

L149-150 – "For the comparison to UCASS observations, CAPS measurements were opted to overlap with the UCASS sampling range from 0.79 to 13.90 $\mu$m in diameter." This sentence is unclear, please reword.

L145-152 – Please add information about the processing applied to the CAS and the CIP. For the CAS, how were refractive index assumptions treated? For the CIP, how was size defined?

L145-152 – how was data in the size overlap region between the CIP and the CAS dealt with? Is one instrument selected in preference to the other?

L154-159 – it is not clear whether UCASS data is being used to evaluate GARRLiC/AERONET/SKYNET or the other way round.

L190 – "aerosol volume concentration" – is this as a column mean?

L201-202 – what resolution is the GDAS meteorological data at?

L208 – "south-western" -> "southwesterly"

L239 – should be 'left' panel?

L 258 – "between 4.5 and 5.5 km height" – this doesn't seem to reflect the figure – the main concentrations seem to be between ~3-5 km, as indicated by the grey shading.

L263 – the figure does not look like an average ratio of 0.92 in the dust plume (3-5km) – with UCASS concentrations around 45 cm-3 and aircraft around 20 cm-3, the ratio appears closer to 0.5.

L255-267 – the authors should discuss the fact that the concentrations from aircraft & UCASS differ significantly above 5km and the possible reasons to explain this. The same behaviour also appears in panel b – in both cases UCASS drops to zero while the aircraft still measures particles.

L268-272 – "While the number concentration exceeds 20 cm$-3$ in this layer, the higher concentrations above 30 cm$-3$ as measured during the first launch were now found only at around 4 km and just below 5 km height. The descent of the larger number concentrations to lower heights is indicative of gravitational settling of particles in the uppermost part of the dust plume during transport." – Fig 4b does not support either of these sentences. While the transition from fig 4a to 4b suggests that overall concentrations have dropped, the heights with concentrations > 30cm-3 appear to have increased from ~3-5 km to ~4.5-5.5 km.

L302 – "between 10 and 50 cm-3" – is this total number concentration or dN/dlogD? If it is the latter, the numbers do not seem to match up with those on the y-axis for the data.

L305-307 – this seems a rather unfair criticism of the data. If one was to draw an envelope around the error bars for the aircraft data, the UCASS data, even for bin 4, would still appear to fall within the uncertainty range.

L310-312 – the data still agree within the aircraft error bars, despite the offset.

L316 – I believe the Liu et al. (2018) effective diameter covers the size range 1 to 20 microns diameter.

L320-322 – and also the different size ranges used to calculate deff, in some cases.

Section 3.4, L324-358:

- Given the fairly large differences between aircraft and UCASS in fig 6a, it would be useful to mention in the previous section that what appear to be small differences in dN/dlogD can translate to very large differences in volume distribution.

- Can the authors comment on where the difference in the ~5-10 micron size range in fig 6a comes from? Would it be the 3-4km layer in fig 5b?

L334-336 – this is a surprising statement – is the UCASS/aircraft comparison from figure 4b not available for a column integrated comparison, given that a vertical comparison is already given in fig 4b?

L 336 – it would be useful to see fig 5 as dV/dlogD as well (see also comment in figure section) to see if the bimodal nature persists vertically, or is featured only in one layer. This may also help explain differences between the remote sensing retrievals and the in-situ observations.

L340-353 – what about contributions from different aerosol types towards the bi-modal distribution?

L372 – While worth retaining the Estelles (2018) reference, the authors may like to cite Kudo et al., (Optimal use of PREDE POM sky radiometer for aerosol, water vapor and ozone retrievals, Atmos. Meas. Tech. Discuss. [preprint], https://doi.org/10.5194/amt-2020-486, in review, 2020.) which publishes some of the same data.

Conclusion

L448-449 – "Furthermore, it was found that the number concentration of large particles decreased with altitude." – This was not evident in the article and results presented. See comments referring to results section of the paper.

Can the authors make any comments or conclusions about differences in results and data comparisons previously published from LOAC, given the differences in instruments described in the introduction? Based on the new results here, are there clear advantages to either LOAC or UCASS that can be shown?

Data availability – please check this statement is in accord with ACP data policy – I do not believe 'available from the authors' is now acceptable.

Figures Fig 1 – caption – presumably the grey bars and UCASS dots are for AOD? The caption however reads like these represent AE – please reword.

Fig 4 – caption – what does the shading around the aircraft data represent? Fig a – why does the distance (black) line disappear abruptly above 5.2km when data is still present for aircraft and UCASS? If CAPS data is only shown up to 14 microns, this comes from the CAS and not the CIP, and this should be corrected in the caption.

Fig 5 – the panels should be enlarged so that the plots are easier to interpret.

Fig 5 – it would be useful to include this figure also as dV/dlogD, so that the discrepancies/similarities between fig 5 and fig 6 can be easily traced.

Fig 7 – enlarge panels to improve readability

Fig 8 – enlarge panels to improve readability, and also enlarge legend font size. In caption, define what a/b/c/d refer to.

[Figure]

---

## Author Comment (AC1) · 12 Mar 2021

AC: We would like to thank the Referees for their comments which helped to improve the quality of our manuscript. Please find our point-by point replies in blue font below.

Please note that we have added Johannes Bühl to the list of co-authors. He operated and maintained the PollyXT instrument during its deployment at Limassol and, thus,

assured consistently high data quality. We have also revised the Acknowledgment to include further people that deserve to be mentioned there.

Anonymous Referee #1

RC: This paper aims to validate a state-of-the-art sonde for dust profiling based on an in-situ optical particle counter approach. The experimental design is really interesting because it can help, for instance, to explore the lowermost troposphere where lidars are typically blind, leading to complement their measurements for modelling applications. The manuscript is very well structured, allowing for a clear comprehension of the research involved. However, I found some issues to be addressed, mainly minor comments.

AC: We thank the Referee for the overall positive assessment of our work.

RC: lines 107-108: It is supposed that the use of Mie scattering has a small effect on the calculated size distributions even in the presence of non-spherical particles. Can you provide a quantification of this?

AC: The following text is added in line 110: "More specifically, Johnson et al. (2011) estimate a maximum error of 21% related to the assumption of spherical dust particles which they assess as moderate compared to the other errors inherent in the derivation of the total optical parameters."

RC: lines 188-189: In the instrumentation section (2.2.) is mentioned that the PollyXT allows for measuring polarized components at 355 and 532 nm. It seems (from the sentence in lines 188-189) that only total signals at these wavelengths are used. Does not GARRLIC use polarized components?

AC: The polarization information is a -relatively- new feature of the GARRLiC retrieval. To be more specific, when we performed the retrievals, it was still in an experimental phase, whereas it has been some time now available for use.

RC: lines 193-199: it would be nice to provide an estimation of the uncertainties of the

GARRLIC derived products.

AC: The following text is added in line 206: "Estimation of the different uncertainties of the GARRLiC derived products is provided in previous works (e.g. Torres et al., 2017), and it has been lately tested for integration in the algorithm (Herrera et al., 2019)." Lines 226-233: How are you sure what is dust and what is not? Taking into account the different origin of air masses shown in figure 3, I recommend to include in fig 1 an additional panel ploting information from depolarization (at least volume linear depol ratio, but particle depol ratio is preferred). This also brings me an additional point. What is the uncertainty for depolarization products? How were those channels calibrated? A detailed description is not needed, some relevant references should be enough.

We thank the Referee for this comment. Based on the suggestion above, we have added a new Figure 2 that shows the profiles of the volume and particle linear depolarization ratio around the times of the five UCASS launches shown in Figure 1. These clearly show the presence of mineral dust over the lidar site. The uncertainty of these products is outlined in Engelmann et al. (2016). Calibrations are performed following EARLINET protocol, i.e. Freudenthaler (2016). The following text has been added to Section 2.2 when listing the measurement capability with respect to the particle linear depolarisation ratio: "The latter parameter is highly sensitive to particle shape and the corresponding measurements are calibrated following the methodology outlined in Freudenthaler (2016)"

In addition, a description of new Figure 2 was added to Section 3.1: "The profiles of the particle linear depolarisation ratio at 532 nm in Figure 2 provide evidence that mineral dust was present over the measurement site and occurred in well-mixed layers. Values larger than 0.20 and as high as 0.33 are generally observed for this particle type (Freudenthaler et al., 2009) and are detected throughout the better part of the aerosol layer while the influence of local aerosols leads to the lower values close to the surface."

Engelmann, R., Kanitz, T., Baars, H., Heese, B., Althausen, D., Skupin, A., Wandinger, U., Komppula, M., Stachlewska, I. S., Amiridis, V., Marinou, E., Mattis, I., Linné, H., and Ansmann, A.: The automated multiwavelength Raman polarization and water-vapor lidar PollyXT: the neXT generation, Atmos. Meas. Tech., 9, 1767–1784, https://doi.org/10.5194/amt-9-1767-2016, 2016.

Freudenthaler, V., Esselborn, M., Wiegner, M., Heese, B., Tesche, M., Ansmann, A., Müller, D., Althausen, D., Wirth, M., Fix, A., Ehret, G., Knippertz, P., Toledano, C., Gasteiger, J., Garhammer, M., and Seefeldner, M.: Depolarization ratio profiling at several wavelengths in pure Saharan dust during SAMUM 2006, Tellus B, 61, 165-179, doi:10.1111/j.1600-0889.2008.00396.x, 2009.

Freudenthaler, V.: About the effects of polarising optics on lidar signals and the Δ90 calibration, Atmos. Meas. Tech., 9, 4181-4255, https://doi.org/10.5194/amt-9-4181-2016, 2016.

RC: Lines 413-414: How did you deal with the incomplete overlap region to estimate AOTs?

AC: We identified the lowermost trustworthy value and extended the profiles to the surface using that value. A corresponding statement was added to Section 2.2: "Lidar-derived values of aerosol optical depth are inferred by extending the profiles down to the surface using the lowermost trustworthy value above the overlap range."

RC: Lines 426-428: Because you used GRASP in this study combining Sun-photometer and PollyXT data, profiles of refractive index and single scattering albedo can be retrieved. This information might support your argument here.

AC: In this paper we focus on UCASS results that can be validated with the help of independent in-situ measurements, i.e. the height-resolved information on particle number concentration and particle size distributions. Nevertheless, we have looked at the findings of the GARRLiC retrieval for measurements during launches 1 and 2 and found

the following results at 532 nm: For launch 1: a real part of 1.48 ± 0.05 and an imaginary part of 0.003 ± 0.02 leading to a single-scattering albedo of 0.89 ± 0.07 For launch 2, a real part of 1.47 ± 0.03 and an imaginary part of 0.006 ± 0.03 leading to a single-scattering albedo of 0.92 ± 0.03

RC: Figure 1: For the sake of clearness, I recommend to use symbols or colors with more contrast for Limassol and Paphos data.

AC: We appreciate the Referee's concern. However, we don't think that it is necessary to provide a large contrast between the data from the two sites. In fact, the purpose of showing data from both sides is to demonstrate that there has been little difference in the aerosol conditions. This allows us to discuss the combined measurements with UCASS launched from Paphos and the Polly lidar based at Limassol. No changes have been made to the figure.

RC: Figure 4: Is there no information about the horizontal distance above roughly 5 km for the case launch 1?

AC: This is now corrected: the black line in what is now Figure 5a was extended to the maximum altitude.

RC: Figure 8: Why do the Klett profiles shown here? It is well-known that this method cannot provide consistent extinction information. Even more, why is the full overlap height different from the extinction and the backscatter profiles shown here?

AC: The figure shows extinction profiles derived by using the Raman method (blue line) and the Klett method (solid and dotted lines). For the comparison to the UCASS measurements and to resolve the same vertical features as identified in the UCASS data (with a height resolution of 5 m), we intended to keep the true vertical resolution of the lidar measurements as fine as possible. As a consequence, the Raman extinction profiles are rather noisy. We therefore also derived Klett extinction profiles using lidar ratios of 40 sr and 60 sr to cover the likely range of values. Indeed, the blue line of the

Raman extinction coefficient mostly stays between the two Klett extinction profiles with a tendency towards the one derived with the lower lidar ratio of 40 sr. The Klett profiles are obtained from the combination of far-field and near-field channels (Engelmann et al., 2016) and, thus, extend further down to the surface. Overall, we believe that the reader gets a better impression of the dust conditions from this comprehensive display of extinction profiles.  

————————————————

[Figure]

**Fig. 1.** Figure 2: Profiles of the linear volume depolarization ratio (black line with grey error range) and the particle linear depolarization ratio (dark green line with light green error range) measured by

[Figure]

**Fig. 2.** Figure 4

[Figure]

**Fig. 3.** Figure 8

---

## Author Comment (AC2) · 12 Mar 2021

RC: General comments: This paper presents new results from a balloon-borne sensor, UCASS, demonstrated and evaluated in dust aerosol during a field campaign in Cyprus. The results demonstrate the usefulness of the instrument and its ability to explore the nature of dust aerosols in the atmosphere. The paper performs a comparison of the instrumental UCASS data against in-situ aircraft observations and ground-based remote sensing. This is a challenging task and the authors present valuable conclusions with regard to the retrieved size distributions, the importance of vertically-resolved information and the ability of UCASS to provide extinction coefficients. The

paper is well-written and well presented, and I recommend publication following the minor corrections listed be- low.

AC: We would like to thank the Referee for their comments which helped to improve the quality of our manuscript.

RC: Title: I suggest removing 'Measurement Report' or rewording it in to the rest of the title – having it at the start of the sentence with a colon implies that it is a particular type of article/publication at ACP, which is not the case.

AC: This is indeed a particular type of article in ACP that has been introduced recently. More information can be found here: https://www.atmospheric-chemistry-and-physics.net/about/manuscript_types.html.

Abstract RC: L1-2 – is this the first publication of UCASS measurements? What is novel about the new measurements/data collected? This should be clearly communicated here or later in the abstract.

AC: Yes, this paper is the first publication of UCASS balloon-borne observations. So far, there is only the technical paper (Smith et al., 2019). Smith, H. R., Ulanowski, Z., Kaye, P. H., Hirst, E., Stanley, W., Kaye, R., Wieser, A., Stopford, C., Kezoudi, M., Girdwood, J., Greenaway, R., and Mackenzie, R.: The Universal Cloud and Aerosol Sounding System (UCASS): a low-cost miniature optical particle counter for use in dropsonde or balloon-borne sounding systems, Atmos. Meas. Tech., 12, 6579–6599, https://doi.org/10.5194/amt-12-6579-2019, 2019.

For clarity, the following statement was added at the end of the introduction: "This is the first paper that is focussed on presenting results of UCASS measurements from field deployment."

RC: L13-14 – "An overestimation of the extinction coefficient of a factor of two was found for layers with particle number concentrations that exceed 25 cm$-3$." An overestimation of lidar or UCASS? Please provide an indication/quantification of the fraction

of layers/instances where this disagreement is found, so the reader can put the differences into context.

AC: The statement was revised to: "An overestimation of the UCASS-derived extinction coefficient of a factor of two compared to the lidar measurement was found for layers with particle number concentrations that exceed 25 cm-3, i.e. in the centre of the dust plume were particle concentrations where highest."

RC: An additional sentence should be given at the end of the abstract to sum up the overall abilities of UCASS and it's potential to provide future measurements and/or insights into environmental data.

AC: The following statement was added at the end of the abstract: "In the future, profile measurements of the particle number concentration and particle size distribution with the UCASS could provide a valuable addition to the measurement capabilities generally used in field experiments that are focussed on the observation of coarse aerosols and clouds."

Introduction RC: Paragraph 1 – Stocker et al. (2013) is missing from the reference list. Citing Stocker twice for this broad paragraph should be avoided – many other papers cover these topics and could be cited here.

AC: New references were added in the first paragraph: Rodriguez et al. (2002); Kaufman et al. (2005); Quaas (2011).

Rodriguez, S., Querol, X., Alastuey, A., and Plana, F., Sources and processes affecting levels and composition of atmospheric aerosol in the western Mediterranean, J. Geophys. Res., 107( D24), 4777, doi:10.1029/2001JD001488, 2002.

Kaufman, Y. J., Boucher, O., Tanré, D., Chin, M., Remer, L. A., and Takemura, T.: Aerosol anthropogenic component estimated from satellite data, Geophys. Res. Lett., 32, L17804, doi:10.1029/2005GL023125, 2005.

Quaas, J.: The soot factor, Nature 471, 456-457, doi:10.1038/471456a, 2011.

RC: L37 – "The main zones of cyclogenesis..." – this sentence is unclear.

AC: An explanatory reference to Bou Karam (2010) related to this term was added in this sentence.

Bou Karam, D., C., Flamant, J., Cuesta, J., Pelon and E. Williams: Dust emission and transport associated with a Saharan depression: February 2007 case, J. Geophys. Res. Atmos., 115 (D4), D00H27, https://doi.org/10.1029/2009JD012390, 2010.

RC: L61 – "with the exception of wing-mounted probes" -> "with the exception of airborne wing-mounted probes"

AC: Corrected, thank you.

RC: L81-82 – are these diameters optical diameters?

AC: Yes, it is clarified.

RC: L100 – please give the particle concentration in cm-3 as this is more interperable and in line with units used in the abstract.

AC: It is now given in cm-3: 3.5 * 103 cm-3.

RC: L101 – Equation $n_i = C_i/V$ – should V be lower case as defined in the previous sentences?

AC: Corrected, thank you.

RC: Section 2.1 – is it possible to retrieve data from the UCASS descent as well as the ascent?

AC: Yes, it is in principle. However, the descent occurs under less controlled conditions than the ascent despite the use of a parachute. As a consequence, the position of the UCASS' opening with respect to the fall direction can have a strong and unquantifiable effect on the OPC's sampling flow rate.

RC: L114 – please confirm that effective diameter is calculated over the full measured

size range (several previous observational studies have selected only part of the size range).

AC: A corresponding sentence was added: "The effective diameter is calculated over the entire UCASS measured size range."

RC: Equation 4 – why is i summed up to the value of 10?

AC: A corresponding sentence is added: "Using the measured number concentration for each UCASS bin (10 bins in total) . . ."

RC: L145 – is this a CIP15?

AC: The CIP part of the used CAPS is a CIP15. But no CIP data is used for this publication.

RC: L148 – insert 'optical diameter' for the CAS

AC: It is corrected.

RC: L149-150 – "For the comparison to UCASS observations, CAPS measurements were opted to overlap with the UCASS sampling range from 0.79 to 13.90 $\mu$m in diameter." This sentence is unclear, please reword.

AC: Reworded to: "For the comparison to UCASS observations, CAPS measurements were opted to fit within the UCASS sampling range spanning from 0.79 to 13.90 $\mu$m in diameter."

RC: L145-152 – Please add information about the processing applied to the CAS and the CIP. For the CAS, how were refractive index assumptions treated? For the CIP, how was size defined?

AC: The CAS size distributions are calculated using Monte Carlo simulations and a set of refractive indices from the literature representative for the aerosol of the sampled airmass as an input (Dollner et al., in preparation). Explaining the entire data processing

of CAS is beyond the scope of this manuscript and is done in a separate publication by Dollner et al. (in preparation). Dollner, M., et al.: "Dust-impacted cirrus clouds during A-LIFE", Atmos. Chem. Phys., in preparation, 2021.

RC: L145-152 – how was data in the size overlap region between the CIP and the CAS dealt with? Is one instrument selected in preference to the other?

AC: No CIP data was used for this publication.

RC: L154-159 – it is not clear whether UCASS data is being used to evaluate GAR-RLiC/AERONET/SKYNET or the other way round.

AC: Our approach was to evaluate UCASS-derived parameters with independent observations of the same parameter whenever this was possible during A-LIFE. This initially referred to the particle number concentration and size distribution as also measured by the research aircraft. After calculating the extinction coefficient profile from the UCASS measurement, we wanted to see if it agrees with the more direct measurement of a Raman lidar. The comparisons to the passive remote sensing retrievals was added as sites with the required measurements are much more abundant than research aircraft deployments or lidar sites. We did not want to miss the opportunity to examine if UCASS and GARRLiC/AERONET/SKYNET could also be reconciled to some degree. This information is also useful for future UCASS deployments that might not be embedded in big field experiments but rather located at sites that are restricted to findings from GARRLiC/AERONET/SKYNET retrievals.

RC: L190 – "aerosol volume concentration" – is this as a column mean?

AC: Yes, it is added in the sentence: "The output of the retrieval provides columnar aerosol volume concentration . . ."

RC: L201-202 – what resolution is the GDAS meteorological data at?

AC: It is added: âĹij50 km resolution

RC: L208 – "south-western" -> "southwesterly"

AC: Corrected.

RC: L239 – should be 'left' panel?

AC: Corrected. There have been some last-minute changes in the figures and the text had not been updated accordingly. Thank you for pointing that out.

RC: L 258 – "between 4.5 and 5.5 km height" – this doesn't seem to reflect the figure – the main concentrations seem to be between âĹij3-5 km, as indicated by the grey shading.

AC: Indeed, it is reworded to: "...shows the highest particle concentrations of more than 35 cm-3 between 3.5 and 4.5 km height."

RC: L263 – the figure does not look like an average ratio of 0.92 in the dust plume (3-5km) – with UCASS concentrations around 45 cm-3 and aircraft around 20 cm-3, the ratio appears closer to 0.5.

AC: This value had not been updated. The correct one is 0.77. It is now corrected in the text.

RC: L255-267 – the authors should discuss the fact that the concentrations from aircraft & UCASS differ significantly above 5km and the possible reasons to explain this. The same behaviour also appears in panel b – in both cases UCASS drops to zero while the aircraft still measures particles.

AC: Figures 2 and 4 (now Figures 3 and 5) show that the horizontal distance between the two platforms was increasing with increasing altitude. Thus, there is a possibility that the two instruments did not sample within the same air mass. During the first launch, the aircraft was sampling south of Paphos, whereas the UCASS was heading eastward. Indeed, the aircraft was sampling upwind and Figure 1 shows that the dust layer got deeper after the first UCASS launch. Consequently, the aircraft could have

detected dust particles at heights above 5 km that were still dust-free in the UCASS measurement.

RC: L268-272 – "While the number concentration exceeds 20 cm−3 in this layer, the higher concentrations above 30 cm−3 as measured during the first launch were now found only at around 4 km and just below 5 km height. The descent of the larger number concentrations to lower heights is indicative of gravitational settling of particles in the uppermost part of the dust plume during transport." – Fig 4b does not support either of these sentences. While the transition from fig 4a to 4b suggests that overall concentrations have dropped, the heights with concentrations > 30cm-3 appear to have increased from âLij3-5 km to âLij4.5-5.5 km.

AC: Thank you for spotting this error. The description of the figure was revised to: "...were now found only at around 4.5 km and between 5.0 and 5.5 km height" and the statement about gravitational settling was removed.

RC: L302 – "between 10 and 50 cm-3" – is this total number concentration or dN/dlogD? If it is the latter, the numbers do not seem to match up with those on the y-axis for the data.

AC: It is dN/dlogD. It is corrected as following: "between 10 and 100 cm-3".

RC: L305-307 – this seems a rather unfair criticism of the data. If one was to draw an envelope around the error bars for the aircraft data, the UCASS data, even for bin 4, would still appear to fall within the uncertainty range.

AC: Yes, it is corrected to: "The particle number concentrations agree within their error bars for the entire size range."

RC: L310-312 – the data still agree within the aircraft error bars, despite the offset.

AC: Yes, it is corrected to: "Despite the offset between the observations in the height layer from 3.6 to 4.6 km, they still agree within the error bars of the aircraft."

RC: L316 – I believe the Liu et al. (2018) effective diameter covers the size range 1 to 20 microns diameter.

AC: Yes, it was corrected: "... gave a mean effective diameter of 4.0 $\mu$m (Ryder et al., 2018) and 5.0 to 6.0 $\mu$m (Liu et al., 2018) for the size ranges from 0.1 to 100.0 ($\mu$m) and from 1.0 to 20.0 ($\mu$m), respectively."

RC: L320-322 – and also the different size ranges used to calculate deff, in some cases.

AC: Yes, it was corrected to: "This is likely due to the fact that the UCASS as deployed during A-LIFE measured only up to particle diameters of 14 $\mu$m, and also due to the different particle size ranges used to calculate effective diameter in some cases."

RC: Section 3.4, L324-358: - Given the fairly large differences between aircraft and UCASS in fig 6a, it would be useful to mention in the previous section that what appear to be small differences in dN/dlogD can translate to very large differences in volume distribution.

AC: Line 315: The following sentence is added: "Please note that small differences in dN/dlogD can translate to very large differences in volume size distribution."

RC: - Can the authors comment on where the difference in the âĹij5-10 micron size range in fig 6a comes from? Would it be the 3-4km layer in fig 5b?

AC: The following text was added in line 341: "The large differences in the volume concentration observed by the aircraft in the size range from 5 to 10 $\mu$m compared to the UCASS can be attributed to the higher number concentration of coarse-mode particles observed by CAPS and shown in Figure 5b."

RC: L334-336 – this is a surprising statement – is the UCASS/aircraft comparison from figure 4b not available for a column integrated comparison, given that a vertical comparison is already given in fig 4b?

AC: Reworded as following: "The first UCASS launch shown in Figure 7a is also the only case for which a column-integrated volume size distribution is available from both remote sensing retrievals and observations aboard the research aircraft."

RC: L 336 – it would be useful to see fig 5 as dV/dlogD as well (see also comment in figure section) to see if the bimodal nature persists vertically, or is featured only in one layer. This may also help explain differences between the remote sensing retrievals and the in-situ observations.

AC: It can be seen in the upper panel of Figure 5 (now Figure 6) that this feature is visible also in dN/dlogD in at least two of the considered layers. We already state in the text that CAPS detects more particles at larger sizes then the UCASS. While we understand the potential value of plotting Figure 5 (now Figure 6) in dV/dlogD, we believe that the current number of figures is sufficient to support our arguments.

RC: L340-353 – what about contributions from different aerosol types towards the bi-modal distribution?

AC: This can also be a possibility. Text was added as following: "The observed bi-modal peak may be caused by the following reasons: (ii) contributions from different aerosol types…"

RC: L372 – While worth retaining the Estelles (2018) reference, the authors may like to cite Kudo et al., (Optimal use of PREDE POM sky radiometer for aerosol, water vapor and ozone retrievals, Atmos. Meas. Tech. Discuss. [preprint], https://doi.org/10.5194/amt-2020-486, in review, 2020.) which publishes some of the same data.

AC: Kudo et al. (2020) has been added to the list of references. Kudo, R., Diémoz, H., Estellés, V., Campanelli, M., Momoi, M., Marenco, F., Ryder, C. L., Ijima, O., Uchiyama, A., Nakashima, K., Yamazaki, A., Nagasawa, R., Ohkawara, N., and Ishida, H.: Optimal use of Prede POM sky radiometer for aerosol, water vapor, and ozone retrievals,

Atmos. Meas. Tech. Discuss. [preprint], https://doi.org/10.5194/amt-2020-486, in review, 2020.

RC: L448-449 – "Furthermore, it was found that the number concentration of large particles decreased with altitude." – This was not evident in the article and results presented. See comments referring to results section of the paper. Can the authors make any comments or conclusions about differences in results and data comparisons previously published from LOAC, given the differences in instruments described in the introduction? Based on the new results here, are there clear advantages to either LOAC or UCASS that can be shown?

AC: We did not have the opportunity to perform coincident measurements with UCASS and LOAC, and therefore think that any attempt to a comparison would be rather speculative. In future, it would be ideal to have combined launches of the two instruments for a fair comparison of their capabilities.

RC: Data availability – please check this statement is in accord with ACP data policy – I do not believe 'available from the authors' is now acceptable.

AC: We would like to thank the Referee for this comment. We have prepared the data from the UCASS measurements to be published as a digital attachment to this paper. The files contain the time and location of the measurement together with the counts per size bin and the total counts. The data availability statement was revised to: "The UCASS measurements used in this study are provided as a digital attachment to this paper. Sun photometer data are accessible through the AERONET portal at https://aeronet.gsfc.nasa.gov/ (last access 03.04.2021). Data of airborne in-situ measurements during A-LIFE are available from B. Weinzierl upon request. Quicklooks of PollyXT lidar measurements can be found at http://polly.tropos.de/ (last access 03.04.2021). For data access, please follow the guidance provided there."

RC: Figures Fig 1 – caption – presumably the grey bars and UCASS dots are for AOD? The caption however reads like these represent AE – please reword.

AC: We are sorry for the confusion. We tried to fit all information in one sentence and didn't realise that this lead to an ambiguous description. The statement was changed to: "Overview of the aerosol conditions at Limassol (34.7°N, 33.0°E) during the 4-day period from 20 to 23 April 2017 in terms of (a) the aerosol optical depth (AOD) as obtained from AERONET sun photometer observations at 500 nm (red) at Limassol (filled dots) and Paphos (open dots), UCASS (532 nm, black circles), and Polly lidar measurements (grey bar spanning the range of values obtained from multiplying the 532-nm backscatter coefficient with 40 sr and 60 sr, respectively) and the Ångström exponent (AE) for the wavelength pair 440/870nm (blue); and (b) the range-corrected signal at 1064 nm as measured with the Polly lidar. Lines and numbers mark the times and locations of UCASS launches. Low backscatter signal (low aerosol concentrations) is shown in blue while very high backscatter signal (dense aerosol layers) is shown in red."

RC: Fig 4 – caption – what does the shading around the aircraft data represent?

AC: The caption has been revised to "…shading marks the standard deviation obtained as median minus 25 percentile and 75 percentile minus median."

RC: Fig a – why does the distance (black) line disappear abruptly above 5.2km when data is still present for aircraft and UCASS? If CAPS data is only shown up to 14 microns, this comes from the CAS and not the CIP, and this should be corrected in the caption.

AC: This is now corrected: the black line in what is now Figure 5a was extended to the maximum altitude). Yes, there is no CIP data used for this publication (added to the caption).

RC: Fig 5 – the panels should be enlarged so that the plots are easier to interpret.

AC: The size of the figure has been increased in the revised manuscript.

RC: Fig 5 – it would be useful to include this figure also as dV/dlogD, so that the

discrepancies/similarities between fig 5 and fig 6 can be easily traced.

AC: Thank you for the suggestion. However, we prefer to keep the number of figures as it is.

RC: Fig 7 – enlarge panels to improve readability

AC: The size of the figure has been increased in the revised manuscript.

RC: Fig 8 – enlarge panels to improve readability, and also enlarge legend font size. In caption, define what a/b/c/d refer to.

AC: The size of the figure has been increased in the revised manuscript. We have also increased the size of the legend and revised the caption to: "Aerosol extinction coefficient profiles from UCASS (red) and the Raman lidar PollyXT at Limassol. Lidar profiles refer to extinction coefficients obtained using the Raman method (blue) or by multiplying the aerosol backscatter coefficient derived without the use of Raman signals with lidar ratios of 40sr (solid black line) and 60sr (dotted black line). Panels refer to the first (a), second (b), third (c), and fourth (d) UCASS launch (see Figure 1 and Table 1)."

Please also note the supplement to this comment:
https://acp.copernicus.org/preprints/acp-2020-977/acp-2020-977-AC2-supplement.zip
* * *
[Figure]

**Fig. 1.** Figure 1

[Figure]

**Fig. 2.** Figure 4

[Figure]

**Fig. 3.** Figure 8